The serotonin blocker Ketanserin reduces coral reef fish Ctenochaetus striatus aggressive behaviour during between-species social interactions

Staubli Virginie 1
Bshary Redouan 1
http://orcid.org/0000-0001-5592-8963 Triki Zegni 1 2 zegni.triki@gmail.com
1 Faculty of Science, University of Neuchatel , Neuchatel , Switzerland
2 Institute of Ecology and Evolution, University of Bern , Bern , Switzerland
Lukas Dieter
Electronic publication date: 2024 Jan 31
Publication date: 2024
Volume: 12
Electronic Location ID: e16858
Received 2023 Nov 8; Accepted 2024 Jan 9
Copyright: © 2024 Staubli et al.
Copyright year: 2024
Copyright holder: Staubli et al.
License: This is an open access article distributed under the terms of the Creative Commons Attribution License, which permits unrestricted use, distribution, reproduction and adaptation in any medium and for any purpose provided that it is properly attributed. For attribution, the original author(s), title, publication source (PeerJ) and either DOI or URL of the article must be cited.
License URL: https://creativecommons.org/licenses/by/4.0/

Keywords: Social behaviour, Marine mutualism, Aggression, Neurotransmitter, Hormone, Mechanism

Funding: Swiss National Science Foundation 310030B_173334/1 and PZ00P3_209020 Financial support was from the Swiss National Science Foundation (grant numbers 310030B_173334/1 to Redouan Bshary and PZ00P3_209020 to Zegni Triki). The funders had no role in study design, data collection and analysis, decision to publish, or preparation of the manuscript.

==============================
A multitude of species engages in social interactions not only with their conspecifics but also with other species. Such interspecific interactions can be either positive, like helping, or negative, like aggressive behaviour. However, the physiological mechanisms of these behaviours remain unclear. Here, we manipulated the serotonin system, a well-known neurohormone for regulating intraspecific aggressive behaviour, to investigate its role in interspecific aggression. We tested whether serotonin blockade affects the aggressive behaviour of a coral reef fish species (Ctenochaetus striatus) that engages in mutualistic interactions with another species, the cleaner fish (Labroides dimidiatus). Although this mutualistic cleaning relationship may appear positive, cleaner fish do not always cooperate and remove ectoparasites from the other coral reef fish (“clients”) but tend to cheat and bite the client’s protective layer of mucus. Client fish thus often apply control mechanisms, like chasing, to deter their cleaner fish partners from cheating. Our findings show that blocking serotonin receptors 5-HT2A and 5-HT2C with ketanserin reduced the client fish’s aggressive behaviour towards cleaner fish, but in the context where the latter did not cheat. These results are evidence of the involvement of serotonin in regulating aggressive behaviour at the between-species social interactions level. Yet, the direction of effect we found here is the opposite of previous findings using a similar experimental set-up and ecological context but with a different client fish species (Scolopsis bilineatus). Together, it suggests that serotonin’s role in aggressive behaviour is complex, and at least in this mutualistic ecological context, its function is species-dependent. This warrants, to some extent, careful interpretations from single-species studies looking into the physiological mechanisms of social behaviour.

Introduction

A behaviour is usually termed ‘social’ if it affects not only the actor’s fitness but also the fitness of one or several recipients (Hamilton, 1964). Social behaviours can have both positive effects, such as grooming and helping, and negative effects, such as competition and aggression (Tinbergen, 1953). The functional definition of social behaviour implies that recipients can be conspecifics as well as heterospecifics (Oliveira & Bshary, 2021). However, such an inclusive approach may be challenged on a mechanistic level unless intra- and interspecific interactions are governed by the same underlying mechanisms (Oliveira & Bshary, 2021). For instance, aggression is an ubiquitous behaviour that can occur frequently within and between species. This behaviour can be either defensive or offensive, and can serve to deter a predator, competing over limited resources such as territory, food, shelter, and mates, but also promote cooperation by preventing partners from defecting (Barnard, 2004; Raihani, Thornton & Bshary, 2012). Therefore, given the wide range of contexts wherein aggression may occur, most chemical messengers acting on the brain can be involved in regulating this behaviour. However, currently we do not know whether the same neural pathways that regulate aggression within a species are also involved in aggression between different species.

Some chemical messengers are better studied in-depth than others for their role in the decision-making of aggression in a multitude of species and across taxa, such as steroid hormones, neuropeptides and neurotransmitters (Wingfield et al., 1998; Ricklefs & Wikelski, 2002; Adkins-Regan, 2005; Reeder & Kramer, 2005). In fact, several of these neuroendocrine systems are phylogenetically ancient and well-preserved across taxa. One of the most ancient systems is that of the serotonin (Azmitia, 1999). Serotonin, or 5-hydroxytryptamine (5-HT), is a monoamine neurotransmitter that acts as a messenger among nerve cells. This neurotransmitter is also considered a hormone and can act through a family of receptors that can have either an inhibitory or excitatory role when activated (Jones et al., 2020). Decades of research on this molecule have shown its imminent role in modulating aggressive behaviour in various species and taxa, like primates, including humans (Kuepper et al., 2010; Larke et al., 2016), reptiles (Larson & Summers, 2001; Summers et al., 2005), birds (Fachinelli et al., 1989; Sperry, Thompson & Wingfield, 2003), fish (Weinberger & Klaper, 2014; Stettler, Antunes & Taborsky, 2021), and insects (Rillich & Stevenson, 2019). Nevertheless, how serotonin modulates aggression and whether it inhibits or enhances this behaviour, especially in a between-species context, still needs to be answered, given the system’s complexity and the broad contexts wherein aggression may occur (Olivier, 2004; Lischinsky & Lin, 2020). Thus, case-by-case experimental studies using excitatory or inhibitory manipulations for targeted serotonin receptors and in well-defined ecological contexts can be a promising approach to unravelling the effects of serotonin on aggressive behaviour.

Here, we manipulated serotonin in an interspecific social interactions context, an ecological context often overlooked in animal studies on aggressive behaviour and its mechanisms (Triki et al., 2017; Oliveira & Bshary, 2021). A suitable study system is that of the mutualistic cleaner fish Labroides dimidiatus and its various coral reef fish partners (“clients”) that engage in iterated cleaning interactions for parasite removal (Losey, 1972). This relationship between cleaner fish and their clients is mutually beneficial, wherein cleaner fish gain food (Grutter, 1999) and client fish get rid of their ectoparasites, eventually boosting their overall health status (Clague et al., 2011; Triki et al., 2016; Ros et al., 2020; Demairé et al., 2020). However, despite the positive aspect of the relationship, conflict may occur as cleaner fish prefer to feed on the client’s protective mucus (Grutter & Bshary, 2003). As a control mechanism to deter their cleaner fish partner from cheating and encourage them to cooperate, client fish have two possible strategies. They either chase the biting partner or switch to another one for their next inspection, causing the loss of a potential food patch for the biting cleaner fish (Bshary & Grutter, 2002; Grutter & Bshary, 2003). This between-species social relationship, where the client-cleaner fish interactions fluctuate between positive and negative, makes it ideal to ask whether serotonin is a potential mediator of aggressive behaviour and, if so, how it affects this behaviour.

At least 14 different serotonin receptor types are being identified in mammals. Fish, in contrast, have three types (5-HT1, 5-HT2 and 5-HT7) with three subtypes of 5-HT1 and two subtypes of 5-HT2, the 5-HT2A and 5-HT2C, that have been identified so far (Norton, Folchert & Bally-Cuif, 2008; Schneider et al., 2012; Mager et al., 2012). Given that the role of serotonin is strongly dependent on the receptor system (see review by Siever, 2008), we opted for the molecule ketanserin, a potent selective antagonist of serotonin receptors in fish that targets 5-HT2A and 5-HT2C receptors (Whitaker et al., 2011). The 5-HT2A and 5-HT2C receptors are known to play significant roles in modulating aggressive behaviour (Popova et al., 2010; Whitaker et al., 2011). Our study builds on previous work by Triki et al. (2017) on the monocle bream (Scolopsis bilineatus). The authors tested the potential involvement of three neurohormonal systems in underpinning S. bilineatus’s aggressive behaviour towards cleaner fish. They manipulated vasotocin (fish homologue of vasopressin), isotocin (fish homologue of oxytocin), and serotonin 5-HT2A and 5-HT2C receptors. Manipulating serotonin receptors with ketanserin emerged as the only treatment that affected S. bilineatus aggressive behaviour with increased attacks towards cleaner fish. Here, we aimed to test the potential role of serotonin in regulating interspecific aggressive behaviour by using a different species, the bristletooth surgeonfish (Ctenochaetus striatus). Using a similar method on another fish species would enable the comparison of findings and determine whether ketanserin acts similarly when exposed to a comparable ecological context. Ultimately, this is crucial to help disentangle the joint and separate roles of serotonin in regulating aggressive behaviour in the marine cleaning mutualism context across species. The set-up consisted in using wild-caught cleaner fish and their C. striatus clients, manipulating ketanserin in the client fish and testing how our treatment affected their aggressive behaviour towards the cleaner fish.

Furthermore, to investigate whether our treatment may have indirectly impacted the overall quality of the cleaner fish cleaning services, we measured the duration of the cleaning interactions, the amount of tactile stimulations and the cheating rate. Tactile stimulations occur when cleaner fish touches the client fish with its pelvic and pectoral fins, which can reduce the client fish’s stress levels (Soares et al., 2011). Cheating events, on the other hand, can be quantified as client fish body jolts caused immediately after the cleaner fish touches the client fish with its mouth, which indicates that the cleaner fish inflicted pain by taking a mucus bite (Bshary & Grutter, 2002). Triki et al. (2017) found that the blockade of serotonin 5-HT2A/2C receptors with ketanserin in S. bilineatus made them more aggressive towards cleaner fish in the absence of cheating events, but did not affect the aggression rates when a cleaner fish indeed cheated nor the quality of the cleaning interactions. Assuming that 5-HT2A and 5-HT2C receptors may have well-conserved functionality across species and context, we expected similar outcomes while testing a different client fish species.

Methods

Field site and animals

The University of Queensland Animal Ethics Committee (AEC) approved this study under permit number CA 2018/08/1222. We collected the data between July and September 2018 at the Lizard Island Research Station (LIRS) on the Great Barrier Reef (14.678436°S, 145.448280°E). Using barrier nets and hand nets, we caught 20 adult client surgeonfish, Ctenochaetus striatus, and 20 adult cleaner fish, Labroides dimidiatus. The sample size was calculated based on the study by Triki et al. (2017). We housed all caught client fish individually in aquaria of 38 × 38.5 × 60 cm (height × width × length) and provided them with a shelter in the form of a polyvinylchloride cylinder (20 cm diameter and 20 cm length) and continuously aerated seawater. Similarly, we housed the cleaner fish individually in aquaria of 37 × 38 × 45 cm with polyvinylchloride pipes as shelter (2 cm diameter and 13 cm length). All client fish came from the same reef site (Osprey), whereas ten cleaner fish were from the reef site Northern Horseshoe and the other ten from Corner Beach. The cleaner fish sites of capture were relevant for another study that used these fish for behavioural and cognitive tests (Triki et al., 2020). This aimed at maximising the reuse of wild animals for ethical reasons (ASAB, 2020).

Immediately after capture, we deparasitized C. striatus using a freshwater bath for 3 min (Jones & Grutter, 2005) followed by an anti-helminthic aerated bath of Praziquantel (ICN Biomedicals Inc., Aurora, OH, USA) (1:100,000) overnight. We performed this step to ensure the welfare of fish in captivity and prevent potential parasitic infections from affecting the cleaner-client fish interaction patterns during behavioural tests. We also measured fish body mass where C. striatus weighed on average 157.18 ± 49.57 grams (mean ± SD), L. dimidiatus from Northern horseshoe weighed 3.69 ± 0.99 grams, and L. dimidiatus from Corner beach weighed 3.25 ± 0.89 grams. We fed daily the C. striatus with a mixture of fish flakes and mashed prawns smeared on a Plexiglass plate and L. dimidiatus with mashed prawns smeared on Plexiglas plates. Furthermore, we allowed all caught fish to acclimate for at least 2 weeks before starting the experiment and monitored them on a daily basis at least twice a day. By the end of the experiment, we returned and released all the surgeonfish C. striatus to their respective site of capture. Half of the cleaner fish L. dimidiatus were released back into their capture site, while the other half had their brains sampled by Triki et al. (2020).

Neuromodulator treatment

Based on the methods by Triki et al. (2017), we prepared the ketanserin treatment by dissolving 20 mg ketanserin (+)− tartrate (S006; Sigma-Aldrich, St. Louis, MO, USA) in 5 mL solution of 95% saline and 5% ethanol. We used the exact dosage and protocol as that by Triki et al. (2017), where we injected intramuscularly C. striatus with 10 μg/g of body weight. In the control condition, we injected the fish with a saline solution of identical volume to the ketanserin solution. All injections were prepared prior to each trial. The client fish were caught using a silicon hand net and placed in a container with seawater. After administering injections, the client fish were immediately transferred to the test tank. The intervention was minor and did not require prior anaesthesia. By following the protocol used by Triki et al. (2017), we were able to compare the effects of a single dose of ketanserin on the aggressive behaviour of client fish towards cleaner fish in a marine cleaning mutualism. Furthermore, we did not record any adverse effects of the treatment, such as lack of appetite, abnormal swimming behaviour, or changes in the skin, eyes and gills.

Experimental set-up

We paired every focal client fish with a partner cleaner fish throughout the experiment on a size-based rule; for instance, the largest client fish formed a pair with the largest cleaner fish, and so forth. The 20 pairs, once formed, did not change during the experiment. In a large round plastic tank measuring 105 cm in diameter and 42 cm high and containing continuously aerated seawater, we placed a pair of cleaner fish and client fish while keeping them separated by an opaque barrier (Fig. 1) and allowed them to acclimate to the experimental tank overnight. The focal fish received their treatment injection (saline or ketanserin) 10 min prior to the start of a trial. The trial started when the experimenter lifted the opaque barrier and allowed the two fishes to interact. We video-recorded these interactions for 15 min. At the end of the trial, both fish were returned to their respective home aquaria. We did not expect or observe any adverse effects of ketanserin treatment. Importantly, we did not feed the cleaner fish before the trials as satiated cleaner fish tend to bite less (Triki et al., 2023).

Figure 1 Schematic representation of the experimental set-up to record the client’s and cleaner fish’s behavioural interactions.

Every client-cleaner fish pair was placed in a round plastic grey tank with a Plexiglas separation between the two fish to allow them to acclimate to the set-up. It is important to mention that fish did not have access to the two small GoPro® placed inside the tank. For that, we have placed a see-through barrier to prevent access. At the top, we installed a third camera, 5-Olympus®, to view the tank differently.

We ran the experiment for five consecutive days and tested eight cleaner-client pairs per day. Given that our experimental set-up is a matched design, we tested every cleaner-client pair twice, once with the ketanserin treatment and once with the control, in a counterbalanced manner with a time gap of two days between the two injections. Using a matched design allowed us to isolate and evaluate the effects of the treatment while controlling for individual variations (Selvin, 2004).

Behavioural analyses

The treatment identity of the video recordings was concealed from the experimenter who encoded these videos by renaming the files with running numbers (#1, #2, and so forth) to avoid potential subconscious bias. The experimenter used the open software CowLog 3.0.2. to extract the following cleaner-client interaction behaviours: (a) the total duration of the cleaning interactions; (b) the total duration of tactile stimulations; (c) the number of client fish’s jolts; and (d) whether the client fish chased the biting cleaner fish after a body jolt, i.e., provoked aggression. Furthermore, there was also chasing behaviour outside the context of the client fish responding to a cheating instance. These aggressions also occurred during cleaning interactions, but we could confirm on the video that no prior mouth contact could have caused the chasing. Therefore, we categorised these chasings as (e) unprovoked aggression.

Data analyses

We used the open-access software R version 4.2.1 (R Core Team, 2022) to run all statistical analyses and generate the figures. Overall, we ran five different statistical models to test whether serotonin manipulation can affect the between-species social interaction patterns of the coral reef fish C. striatus. The first model was a Linear Mixed Effects model (LMER), where we fitted the duration of cleaning interactions as the response variable, treatment (ketanserin vs saline) as a categorical predictor variable, and the client-cleaner fish pair identity as a random factor because every client-cleaner fish pair was tested twice, once when the client fish was injected with ketanserin and once with saline. Given that cleaner fish were caught from two different reef sites, we also added the reef site identity as a random factor. The second model was also an LMER model, with the duration of tactile stimulation as the response variable, treatment as a predictor, and standardised and log-transformed duration of cleaning interactions as a co-variate, while client-cleaner pair and cleaner fish reef site identities as random factors. The third model investigating client fish body jolts is a Generalized Additive Model for Location Scale and Shape (GAMLSS) because GAMLSS models support a wide range of data distribution than other statistical models (Stasinopoulos & Rigby, 2008). For instance, our model contained data that followed negative binomial distribution but contained several zeros. Therefore, we fitted a GAMLSS model with zero-inflated negative binomial error distribution for the count of client fish jolts as the response variable, treatment as a predictor, standardised and log-transformed duration of cleaning interactions as a co-variate, while client-cleaner pairs and cleaner fish reef site identities as random factors. Fourth, we fitted a GAMLSS with beta-inflated error distribution (data range was [0, 1]) since our response variable in this model was the proportion of provoked aggressions calculated from total jolts. The predictor variable was treatment, and client-cleaner pair and cleaner fish reef site identities were random factors. The final model was also a GAMLSS, but with zero-inflated Poisson distribution, wherein we fitted the count of unprovoked aggressions as the response variable, treatment as the predictor, and standardised and log-transformed duration of cleaning interactions as a co-variate, while client fish and cleaner fish reef site identities as random factors. We ran visual and statistical diagnostics (e.g., residual plots, quantile-quantile plots, worm plots, and Lilliefors (Kolmogorov-Smirnov) test for normality) for all five models, and all of them fitted their respective assumptions, such as normality of residuals and homogeneity of variance. Please refer to our step-by-step code provided along with the data via the shared link in the Data and Code accessibility statement for further details. All the collected data were included in the statistical analyses.

Results

We recorded the behaviour of N = 20 individual client fish (Ctenochaetus striatus) under two conditions: ketanserin and saline treatments. This resulted in a total of 40 video recordings. However, two fish did not interact with their partner cleaner fish during the saline treatment, leaving us with 38 video recordings showing cleaning interactions. Our data analyses of the cleaning interaction patterns showed that our neuromodulator treatment did not affect the overall duration of cleaning interactions (N = 38 videos; estimate β [low – high 95% credible interval] = −14 [−90 to 62.1], p-value = 0.703, Fig. 2A), the amount of tactile stimulations received (N = 38 videos; β = 15.2 [−4.67 to 35.1], p = 0.125, Fig. 2B), or the frequency of being cheated by their partner cleaner fish (N = 38 videos; β = −0.432 [−2.32 to 1.30], p = 0.479, Fig. 2C) (Table 1).

Figure 2 Cleaner-client cleaning interaction patterns as a function of neuromodulator treatment.

Boxplots of median, interquartile and ranges of (A) total duration of the cleaning interactions per 15 min of observation; (B) the amount of tactile stimulation provided during these cleaning interactions, here, depicted as a time proportion measure for visualisation (see statistics for details about the analyses per se); and (C) the number of client fish body jolts occurring per 1 s of interaction time (here as well we show the jolt measure as a rate for simplified visualisation (see statistics for details about the analyses per se)). Also, the raw data points are depicted where every matched trial (individual client fish) is connected with a dashed line to visualise individual behavioural changes as a function of the neuromodulator treatment.

Table 1 Summary table.

Summary table with statistical outcomes from the different models testing changes in client fish behaviour as a function of the hormonal treatment manipulation where N = 20 client fish were injected with ketanserin and saline (as a control) in a matched design. The statistical significance level was set at alpha ≤ 0.05. Statistically significant outcomes are indicated in bold. Model name abbreviations are LMER, linear mixed effects model; and GAMLSS, generalized additive models for location scale and shape. Please refer to the main text for a detailed description of the measured behaviours.

Statistical model class (distribution)	Measured behaviour (response variable)	Treatment (ketanserin vs saline)	Covariate (Duration of cleaning interactions)	Explained variance (marginal and conditional R2 or pseudo-R2)	
estimate	95% Confidence level [low – high]	p-value	Estimate	95 % CI	p-value	
LMER (gaussian)	Duration of cleaning interactions	−14.0	[−90.0 to 62.1]	0.703	–	–	–	0.01–0.77	
LMER (gaussian)	Duration of tactile stimulations	15.2	[−4.67 to 35.1]	0.125	52.1	[30.1–74.0]	<0.001	0.41–0.88	
GAMLSS (Zero Inflated Negative Binomial)	Number of jolts	−0.432	[−2.32 to 1.30]	0.479	1.45	[4.49 to −1.59]	0.357	0.20	
GAMLSS (Beta inflated distribution)	Provoked aggression (chasing the cleaner fish after a cheating event)	−0.789	[−2.29 to 0.71]	0.318	–	–	–	0.04	
GAMLSS (Zero Inflated Poisson)	Unprovoked aggression (aggression towards cleaner fish without not preceded by a cheating event)	−0.774	[−1.30 to −0.248]	0.006	−0.867	[−1.27 to −0.467]	<0.001	0.36	

Notably, the client fish provoked aggression towards cleaner fish in the form of chasing the cleaner fish after a cheating event—often indicated by the client fish body jolt—was not affected by the neuromodulator treatment (N = 20 videos—as only 20 videos out of 38 had client fish showing at least one jolt event during the recording; β = −0.789 [−2.29 to 0.71], p = 0.318, Fig. 3a). Importantly, client fish aggressive behaviour in an unprovoked context, when they chase cleaner fish without the latter causing client fish to jolt, was significantly affected by the neuromodulator treatment (N = 38 videos; β = −0.774 [−1.30 to −0.248], p = 0.006, Fig. 3B) (Table 1). The data showed that client fish receiving ketanserin injections were less likely to chase the cleaner fish compared to the control, in the context where such chasing was unprovoked by the cleaner fish.

Figure 3 Client fish’s aggressive behaviour towards the cleaner fish as a function of neuromodulator treatment.

Boxplots of median, interquartile and ranges of (A) provoked aggression, which is the proportion of client fish chasing their cleaner fish partner after a body jolt; and (B) unprovoked aggression occurring per 1 s of interaction time (here we show the unprovoked aggression measure as a rate for simplified visualisation (see statistics for details about the analyses per se)). Also, the raw data points are depicted where every matched trial (individual client fish) is connected with a dashed line to visualise individual behavioural changes as a function of the neuromodulator treatment. Several client fish did not show provoked aggression (see Results).

Discussion

We aimed to test whether the selective blockade of serotonin receptors, the 5-HT2A and 5-HT2C—known for their role in modulating aggression (Popova et al., 2010; Whitaker et al., 2011), affect coral reef fish aggressive behaviour in a between-species social interaction context. Although serotonin manipulation did not change the overall response of C. striatus to cheating events where most of their aggressive behaviour is expected to occur, it affected their “unprovoked” aggression, the rate by which they chased their cleaner fish partner without the latter having caused or provoked such attacks during the cleaning interactions. We found that the serotonin antagonist, ketanserin, reduced the rate of these unprovoked aggressive attacks, suggesting that in the client fish species C. striatus, serotonin action through the 5-HT2A/2C receptors reduces tolerance of non-conspecifics.

In the study by Triki et al. (2017), they injected the client fish species S. bilineatus with ketanserin and compared their behaviour to those injected with saline as a control. In both Triki et al. (2017) and the current study, ketanserin did not affect the client-cleaner fish interaction quality or the client’s response to cheating. This direct comparison of the two very common client reef fishes (Grutter, 1995) belonging to two phylogenetically distant orders within the percomorph radiation (Grutter & Poulin, 1998; Hughes et al., 2018) suggests that serotonin, via 5-HT2A/2C receptors, does not modulate key aspects like client fish’s willingness to receive cleaning services or exerting punishment-like behaviours when their cleaner fish partner defects. In contrast, ketanserin increased the unprovoked aggression rate of S. bilineatus towards cleaner fish, while decreasing it in C. striatus. However, in both studies, the jolt rate as an indicator of cheating was relatively low—here only half of the video observations showed at least one jolt rate. Although cleaner fish in our experiment, as in that of Triki et al. (2017), did not receive food prior to the trials to increase the chances of biting (Triki et al., 2023), they tended to cooperate and rarely cheated their clients. Having limited data to test how serotonin manipulation affects client fish response to cheating warrants careful interpretation. It is unclear whether serotonin does not regulate this behaviour in particular, or we cannot see an effect due to insufficient data.

For the unprovoked aggressive behaviour, our findings in reference to the study by Triki et al. (2017) show evident opposite effects of serotonin manipulation. Although unaware of the exact mechanisms yielding such results, we suggest the following potential explanations. The role of serotonin varies between species, despite the fact that the system itself is ancient and well-preserved across invertebrate and vertebrate species (Azmitia, 1999). If the increase of serotonin decreases aggression in species A, for instance, it might have adopted an opposite role in species B. For example, in rainbow trout (Oncorhynchus mykiss), an increase in serotonin through dietary manipulation had an inhibitory effect on their aggressive response to an intruder (Winberg, Øverli & Lepage, 2001). Similarly, in several reptile species, increased serotonin levels in the brain are accompanied by enhanced aggression (Summers & Greenberg, 1995; Matter, Ronan & Summers, 1998). In other species, like pigeons (Columba livia) and song sparrows (Melospiza melodia morphna), increased serotonin yields lower aggression instead (Fachinelli et al., 1989; Sperry, Thompson & Wingfield, 2003). Even within the same clade, we can see how serotonin mediates aggression in opposite directions. As such, in coral reef fish, the bluehead wrasse (Thalassoma bifasciatum) fluoxetine treatment—a selective serotonin reuptake inhibitor that yields higher serotonin activity—reduces chasing rates towards intruding conspecifics (Perreault, Semsar & Godwin, 2003), while in another coral reef fish species, the bicolour damselfish (Pomacentrus partitus) higher serotonin activity correlates positively with the number of aggressive attacks (Winberg, Myrberg & Nilsson, 1996). Thus, a logical explanation of our findings is that serotonin functionality in C. striatus differs from S. bilineatus. Therefore, serotonin’s role in regulating aggressive behaviour towards heterospecific can be complex, and warrants careful interpretations from single-species studies.

Following the blockade of the 5-HT2A/2C receptors, serotonin might have decreased C. striatus chasing attacks by becoming more available at the synaptic level to bind with and activate other sets of receptors like the 5-HT1A. This receptor is well-known for its role in reducing intraspecific aggression, especially in fish, like bluehead wrasse, T. bifasciatum, the fighting fish Betta splendens, and even in the cleaner fish L. dimidiatus, where excitation or inhibition of this receptor decreases and increases aggressive behaviour towards conspecifics, respectively (Perreault, Semsar & Godwin, 2003; Clotfelter et al., 2007; Paula et al., 2015). Moreover, we know that serotonin receptors can vary in their amount and distribution throughout the central nervous system, Olivier (2004). Using ketanserin as an antagonist in itself might have had further effects. Apparently, the molecular structure of ketanserin can affect its affinity to the 5-HT2 receptors (Herndon et al., 1992; Egan et al., 2000). It is thus possible that quantitative variability in serotonin receptors or receptor affinity to ketanserin between S. bilineatus and C. striatus drove the documented differences. Only extreme measures, like invasive methods involving killing the test subjects and performing immunoreactivity analyses (Frankenhuis-van den Heuvel & Nieuwenhuys, 1984), would show the amount and distribution of the serotonin receptors throughout the brain and how serotonin agonists and antagonists affect their target receptors as well as the indirect effects on the other non-target serotonin receptors.

What adds to the complex role of serotonin in aggressive behaviour is the dose-dependent effect. It should be noted that there was not a consistent reduction in aggression in C. striatus after injections with ketanserin. While this could be due to random data variation, it is also possible that certain individuals responded differently to the dosage given. Serotonin, among other neuromodulators, has an inverted U-shape function (Olivier, 2004; Cano-Colino et al., 2014; Stettler, Antunes & Taborsky, 2021). Therefore, the increase or decrease in these neuromodulators can yield similar behavioural outputs if they are at the two ends of the inverted U-shaped curve. S. bilineatus and C. striatus might have responded differently to the same dose of ketanserin if their dose-dependent curve functions were not similar. Experimentally testing for this dose-dependent effect can help investigate species differences in serotonin function. However, such approach requires significant logistics that pose enormous practical challenges, especially for studies using wild animals.

Finally, our study focused only on the role of serotonin and a specific subset of its receptors in regulating aggressive behaviour. However, several other neuroendocrine systems also regulate aggression, and complex interactions may exist among them. These interactions may explain why ketanserin has different effects on C. striatus and S. bilineatus. Therefore, further research is required to explore the role of other systems, such as testosterone, cortisol (but see Ros, Vullioud & Bshary, 2012), and dopamine (Lischinsky & Lin, 2020), in mediating aggressive behaviour within the cleaning mutualism context, both jointly and separately.

Despite the opposing effects of ketanserin on the client fish’s unprovoked aggression towards cooperative cleaner fish, we conclude that serotonin does affect the client’s tolerance levels to proximity by cleaner fish, similar to what has been described in the literature about serotonin’s role in conspecifics’ tolerance and close contact (Insel & Winslow, 1998; Stettler, Antunes & Taborsky, 2021). This suggests that the serotonin system may have been co-opted to regulate aggression not only when interacting with individuals of the same species but also of different species. The results thus contribute to the still relatively scarce evidence that interspecific interactions may be labelled as ‘social’ not only from a functional but also from a mechanistic perspective (Oliveira & Bshary, 2021).

Supplemental Information

Supplemental Information 1 ARRIVE Checklist.

Click here for additional data file.

We thank the LIRS directors and staff for their support and friendship. A preprint version the article is available in the bioRxiv repository (Staubli, Bshary & Triki 2023).

Additional Information and Declarations

Competing Interests

Author Contributions

Animal Ethics

Data Availability

The authors declare that they have no competing interests.

Virginie Staubli performed the experiments, analyzed the data, authored or reviewed drafts of the article, and approved the final draft.

Redouan Bshary conceived and designed the experiments, authored or reviewed drafts of the article, and approved the final draft.

Zegni Triki conceived and designed the experiments, analyzed the data, prepared figures and/or tables, authored or reviewed drafts of the article, and approved the final draft.

The following information was supplied relating to ethical approvals (i.e., approving body and any reference numbers):

The University of Queensland Animal Ethics Committee (AEC) approved the study under permit number: CA 2018/08/1222.

The following information was supplied regarding data availability:

The data is available at figshare: Staubli, Virginie; Bshary, Redouan; Triki, Zegni (2024). Data from: The serotonin blocker Ketanserin reduces coral reef fish Ctenochaetus striatus aggressive behaviour during between-species social interactions. figshare. Dataset. https://doi.org/10.6084/m9.figshare.11835690.v1.

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
