# Peer review of "The serotonin blocker Ketanserin reduces coral reef fish Ctenochaetus striatus aggressive behaviour during between-species social interactions"

_PeerJ, doi:10.7717/peerj.16858_

## Round 0.1 · original submission · Minor Revisions

Your article entitled “The serotonin blocker Ketanserin reduces coral reef fish Ctenochaetus striatus aggressive behaviour during between-species social interactions” has now been been seen by three reviewers and the reviewers’ comments are appended below.

I share the reviewers' view that this manuscript provides new insights into the hormonal mechanisms underlying aggressive behavior in interspecies interactions.

The reviewers only have sets of minor comments asking for additional explanations. I think adding this information will be helpful to provide context to the findings and make the manuscript more accessible to readers not familiar with. I am looking forward to a revised manuscript.

Reviewer 1 ·

Basic reporting

This is a reasonably well written manuscript, with a clear narrative and a good grasp of the relevant background. The statements made are appropriately supported by the literature references, and the conclusions drawn are supported by the data. The structure is intuitive, and all the data is included in figures, tables, and the supplementary inclusion of the raw dataset and all relevant code. I have noticed some typos and some places where the exposition could possibly be improved. I would also like to see some more information about the comparison between the species used by this study and that of Triki et al., 2017 (see below for specific comments):
Line 36: consider changing ‘the’ to ‘its’
Lines 103-107: consider rewriting
Line 114: rewrite as administering ketanserin or manipulating the activity of these specific serotonin receptors, as ketanserin is not an intrinsic molecule.
Line 133: this instead of theis (typo)
Line 157: did the fish sampled originate from both sites? Was the split 50/50 or was there a skew? It may be helpful to explain that these samples were collected to be used to a different study, as they are not mentioned elsewhere in the manuscript.
Line 239: consider replacing neither with not
Line 242: replace nor for or (if making the change suggested above in line 239)
Lines 266-269: I would suggest removing these two sentences altogether
Lines 271-284: How closely related are the two client fish species? Would we expect the neuromodulatory systems to have evolved in a similar manner? Have any studies looked at differences in expression of these receptors between the two species? Also, would diet restriction of the cleaner fish prior to the experiment increase their motivation to cheat?
Lines 288-290: consider rewriting
Line 292: please give the species name for rainbow trout
Line 296: please give the species names

Experimental design

This manuscript falls well within the scope of the journal. The research question is well defined and meaningful, and the study design is very well thought out. The text addresses all possible ethical concerns, and the methods are very detailed and well written. I particularly like the very clear explanation of the statistical approach, and how it addresses any possible major issues.

Validity of the findings

All conclusions are well stated, linked to the research question and supported by the results. All the analyses carried out are appropriate and sound.

Additional comments

On the ARRIVE checklist, point 2a, please explain where in the text this point is addressed (this point has been addressed by the authors in the manuscript but maybe should be included in the checklist too).

I enjoyed reviewing this manuscipt very much!

·

Basic reporting

The text is well-written and brings clear and technically correct information. The article includes sufficient introduction and background to demonstrate how the work fits into the broader field of knowledge. The aims are clearly stated and preceded by adequate rationale. Figures and tables are relevant to the content of the article, of sufficient resolution, and appropriately described and labeled.

Experimental design

The experimental design is adequate to answer the main question of the study; the variables tested are evident, and the data analysis is all alright.

Validity of the findings

Results are robust and conclusions are supported by the data.

Additional comments

The study presented here tested the possible involvement of serotoninergic mechanisms underlying intraspecific aggression between clients and cleaner fish. To do this, the authors blockade serotonin receptors by using ketanserine on the client fish and evaluated its aggressive behavior towards the cleaning fish. The authors found an effect of serotonin blockade on the client’s unprovoked aggression. The results brought discussion about interspecific differences in the control of similar behaviors in different species.
As usual for this research team, the study is excellent, well conducted, well presented and brings news to the area. The objective is evident, the methods are adequate, as are the results and discussion. I suggest that the article can be accepted with minor revision. I just asked for some explanations which are described below. Congratulations to the team for such an interesting work.

L. 46 – That is probably a mistake. This book is from Niko Tinbergen, and the last print I think was in 1990. Google Booking refers to the book with J Tinbergen (an economist) as an author and shows 2012 as the year of publication. Please. Check this and correct it accordingly.
L. 50 – I think your focus is neuroendocrine mechanism. Consider deleting this sentence “Here, our focus is on physiological mechanisms such as endocrine systems.” and change the following to “For instance, a well-studied social behavior and its underlying neuroendocrine mechanisms is aggression.”
L. 61 – The same. Change the word endocrine to neuro-endocrine to fit your previous sentences. Particularly to talk about serotonin.
L. 65. Please, provide a reference regarding serotonin being considered a hormone as well as regarding the inhibitory and excitatory receptors.
L. 71-73 – I am sure that lots of new studies were published after Olivier 2004. I recommend you cite a more recent work to assert that the mechanism underlying aggression needs to be understood.
L. 147 – A paragraph could be inserted to give the readers a breath.
L. 148 – Did you remove the parasites from the client fish to stimulate cleaners to take mucous?
L. 168-169 and L. 181 – What did you consider as “adverse effect”? A brief explanation would make this assertion more precise.
L. 195 – This is the first time you talk about tactile stimulation. I think it is obvious for readers used to your study. However, I suggest you briefly explain what it means to the fish interaction.
Discussion
It would be interesting to discuss that the control mechanism of interspecific social aggression may involve more than one neuropeptide. A set of monoamines, such as dopamine and oxytocin, can also participate in the control of interspecific aggression. Although it is adequate to start studying a single factor, I think it is important to mention the complexity of this process.

·

Basic reporting

English is good, but could use some rewording to make more concise in many places.
Literature references are sufficient, there was one note in the introduction where an additional citation is warranted.
Data was shared and R code makes sense given the statistics ran.
Hypothesis/Question was given along with a prediction.

Experimental design

Even though the experimental design is close to another publication, the authors do explain how the paper is different and is helping to fill a gap regarding species specific differences in how serotonin modulates different social behaviors such as aggression.
I do make a note below that the readers should not have to refer to another paper for get methods for replication and there are thus a few places where clarification is warranted.

Validity of the findings

I think the benefit to the literature needs to be more clearly hashed out within the discussion. I think the materials are there, but the clarity of the discussion and the conclusion could use some work.
Data were provided.

Additional comments

The authors use ketanserin as a mode to block serotonin receptors in a client fish species to see how that species responds to cleaning events by a cleaner reef fish species. Their overall methods and experimentation closely match another study with the caveat that they use another species. The authors do a decent job explaining the purpose of repeating the study with a second species but could expand on this more in the discussion. The manuscript could use some tidying up in terms of conciseness and clearness of sentences. The authors have many floating “it” statements that can be eliminated with careful rewording of sentences.
Introduction
Line 45: giving a specific behavior aside from “helping” would be more descriptive here
Line 48: floating “it” statement
Line 50: the transition in this opening paragraph from talking about social behaviors to focusing on the endocrine system is rough, another sentence or two within this paragraph would set up why the endocrine system is the system of focus.
Line 52: floating “it” statement
Lines 52-54: Need to rework this sentence to have parallel structure, reads very clunky as is.
Line 66: floating “it” statement
Line 67: floating “it” statement
Line 71: floating “it” statement
Lines 78-79: need citations after the opening sentence.
Line 109: What is “this” referring to?
Line 129: An additional sentence at the end as to why finding similar outcomes between species would be beneficial would be helpful here.
Methods
When talking about the models ran in the data analysis section, the text could be made more concise by stating that the cleaning interactions were standardized, and log transformed only once. All models also have client-cleaner pair and reef fish ID as a random factor which could just be stated once.
Line 144: floating “it” statement
Lines 163-165: The reader shouldn’t have to refer to Triki et al. 2017 to replicate this experiment. More explanation is needed on why 10 ug/g for the dosage and how the fish were handled without anesthesia and injected with ketanserin.
Lines 178 and 180: should spell out minutes.
Line 180: Why was video interaction only done between the fishes for 15 minutes?
Line 187-188: Sentence reads awkwardly, I think you need an “us” between allows and to
Line 208: What is “this” referring to?


Results
If you are going to include individual behaviors on the figures, then they should be noted in the results section. Could these be sex differences? Size differences? Reef differences (I know you used reef ID as a random effect, but visualizing the different reefs on the graph might show interesting trends). Need to expand on what the reader might be interested in looking at in Table 1 instead of just saying that further statistical outcomes are available.
Discussion
Overall, the discussion talks about the results considering previous research. I think the authors could expand some discussion on how the results relate to the ecological role the species play within the environment. Why do we want to study these mutualistic interactions? What can they tell us about the biodiversity of the system or how the system functions? I was again looking for some discussion of the variation between individuals. There could be interesting trends within your data that are not being explored.
There is some discussion about how cleaner fish in the experimental setup tended to cooperate and rarely bit their clients. Could this interaction be because the pairings were not switched up within the experiment? The introduction talked about cleaner fish not biting so the client sticks around, if there is only one client is the cleaner fish likely to “behave” more than a fish in a natural setting with multiple clients?
I was looking for a paragraph that talked about the limitations of the data because the experiments were as natural as possible but with some caveats. How many fish do the cleaner fish typically interact with on a given day? Does feeding the cleaner fish have an impact on how many “bites” are taken from the mucus layer? There was a lot of individual variation within these treatments, could a training period have been used to collect cleaner fish that show more biting tendencies to get at the research question?
The conclusion paragraph is underwhelming, how does this research help us understand biotic relationships of the mutualisms between cleaners and their clients. There is talk about the logistics of using wild animals, so is there another option that could be used instead? I feel like the conclusion paragraph ends on a sour note instead of a “the research showed us X” note.
Line 261: floating “it” statement
Line 278: three floating “it” statements
Line 282: floating “it” statement
Line 290: floating “it” statement
Line 291: floating “it” statement
Line 292: do not need sentence “Examples from the literature support this point.”
Line 305: floating “it” statement
Line 319: floating “it” statement
Line 328: floating “it” statement
Line 329: floating “it” statement
Line 337: what is “this” referring to?

---

## Round 0.2 · accepted · Accept

Thank you for addressing all comments so diligently and for making the respective changes in the manuscript. The additional information helps to clarify how your findings relate to previous studies and how others could potentially build on them. I only noticed a few remaining issues that you might want to address in the final manuscript. I think these edits (edit figure legends, remove one word, update link) can occur during the final proof checks:

- there was one comment from reviewer 3 about the figures, which I realise now was slightly unclear. I wonder whether part of what this reviewer might have been referring to is the seeming difference in the level of data you used during the analyses versus the level of data you used during plotting. For the analyses, you state that you treated each video as a sample, using random factors to nest these within individuals (and other factors). For example, for the analyses of provoked aggression, you list an "N=20 videos" in the result section. However, the corresponding Figure 3a talks about "individual client fish" and the way the data are displayed makes it seem that there are fewer observations. Maybe there is a way to clarify in the figure that there are multiple observations where both during the treatment and the control no aggression was observed. Alternatively it would be worth mentioning this in the legend. In addition, it is not always clear whether you only used matched observations (where clients and partners were observed both in the treatment and the control condition). The figures seem to suggest this, but the first sentence of the result section states otherwise. Again, it would be helpful to briefly clarify this, for example in the figure legends. This is simply about reducing any potential confusion that might arise, as you provide the full code the actual way you analysed the data are clear.

- in the new sentence you added in line 189 about the reuse of the animals, the use of the word "evidently" is not immediately clear because the experience of other experiments can sometimes change the behavior of animals in ways that can make interpretation more difficult. I would suggest removing the word.

- the updated link to the repository that uses the DOI seems broken. The link you originally provided pointing directly to Figshare still works.